# Molnupiravir Revisited—Critical Assessment of Studies in Animal Models of COVID-19

**DOI:** 10.3390/v15112151

**Published:** 2023-10-25

**Authors:** Henrik Berg Rasmussen, Peter Riis Hansen

**Affiliations:** 1Institute of Biological Psychiatry, Mental Health Centre Sct. Hans, 4000 Roskilde, Denmark; 2Department of Science and Environment, Roskilde University, 4000 Roskilde, Denmark; 3Department of Cardiology, Herlev and Gentofte Hospital, Copenhagen University Hospital, 2900 Hellerup, Denmark; peter.riis.hansen@regionh.dk; 4Department of Clinical Medicine, Faculty of Health and Medical Sciences, University of Copenhagen, 2200 Copenhagen, Denmark

**Keywords:** molnupiravir, COVID-19, animal models, antiviral efficacy, animal-to-human extrapolation

## Abstract

Molnupiravir, a prodrug known for its broad antiviral activity, has demonstrated efficacy in animal models of COVID-19, prompting clinical trials, in which initial results indicated a significant effect against the disease. However, subsequent clinical studies did not confirm these findings, leading to the refusal of molnupiravir for permanent market authorization in many countries. This report critically assessed 22 studies published in 18 reports that investigated the efficacy of molnupiravir in animal models of COVID-19, with the purpose of determining how well the design of these models informed human studies. We found that the administered doses of molnupiravir in most studies involving animal COVID-19 models were disproportionately higher than the dose recommended for human use. Specifically, when adjusted for body surface area, over half of the doses of molnupiravir used in the animal studies exceeded twice the human dose. Direct comparison of reported drug exposure across species after oral administration of molnupiravir indicated that the antiviral efficacy of the dose recommended for human use was underestimated in some animal models and overestimated in others. Frequently, molnupiravir was given prophylactically or shortly after SARS-CoV-2 inoculation in these models, in contrast to clinical trials where such timing is not consistently achieved. Furthermore, the recommended five-day treatment duration for humans was exceeded in several animal studies. Collectively, we suggest that design elements in the animal studies under examination contributed to a preference favoring molnupiravir, and thus inflated expectations for its efficacy against COVID-19. Addressing these elements may offer strategies to enhance the clinical efficacy of molnupiravir for the treatment of COVID-19. Such strategies include dose increment, early treatment initiation, administration by inhalation, and use of the drug in antiviral combination therapy.

## 1. Introduction

Molnupiravir is the isopropylester derivative of the nucleoside analog β-D-N4-hydroxycytidine (NHC) [1,2]. This nucleoside analog possesses broad antiviral activity and initially garnered attention as an agent for the treatment of infections caused by the equine encephalitis viruses and other Alphavirus infections [3,4,5,6]. Due to low oral bioavailability in non-human primates, NHC was redesigned as the ester prodrug molnupiravir, which entered preclinical testing for the treatment of seasonal influenza [4,7]. However, with the emergence of coronavirus disease 2019 (COVID-19), molnupiravir was repurposed to treat this new pandemic [4]. 

Findings of significant in vitro anti-SARS-CoV-2 activity of molnupiravir led to preclinical efficacy studies of the drug in animal models of COVID-19 [2]. Studies in mice, Syrian hamsters, Roborovski dwarf hamsters, ferrets, and nonhuman primates indicated substantial in vivo activity of the drug against SARS-CoV-2, as evidenced by effective viral clearance, improved clinical signs, and reduced lung pathology [2]. Also, molnupiravir effectively blocked viral transmission in the ferret model of COVID-19, a model that is characterized by the robust upper airway replication of SARS-CoV-2 while expressing only mild symptoms, and therefore imitating infections with the virus in children and teenagers [8]. The assessment of the efficacy of molnupiravir in the macaque model of COVID-19 was partially hampered by mild clinical disease, but the agent reduced viral load in the airways of the animals [9,10]. 

During the COVID-19 pandemic, molnupiravir entered clinical trials to assess its efficacy against the disease. On 25 October 2021, the European Medicines Agency (EMA) initiated a rolling review of the agent based on results from preclinical studies and the MOVe-OUT trial, a randomized placebo-controlled clinical trial with 716 patients assigned to the molnupiravir group and 717 to the placebo group [11]. A few months later, on 23 December 2021, the United States Food and Drug Administration issued an emergency use authorization for molnupiravir [2]. Likewise, molnupiravir received approval for emergency use in countries such as Japan, and the EMA advised member countries to use molnupiravir for the treatment of COVID-19 while conducting its rolling review [12]. Subsequently, data from additional clinical trials became available that did not confirm the substantial benefits reported by the MOVe-OUT trial, including PANORAMIC, consisting of 12,821 patients assigned to treatment with molnupiravir and 12,962 to placebo [11,13,14], and on 23 February 2023, the EMA recommended against the marketing authorization of the agent [15]. Of note, an observational study, based on US Veterans with immunocompromised conditions and documented SARS-CoV-2 infection, found lower hospitalization or mortality rates in 154 subjects treated with molnupiravir compared to control subjects who did not receive this treatment [16]. However, in a larger observational study of 1459 matched pairs of US Veterans with a history of COVID-19, where one subject in each pair initiated molnupiravir treatment within 3 days of diagnosis, a reduction in hospitalization or mortality was not observed in the molnupiravir-treated subjects, except for a subgroup presenting without symptoms [17]. 

Early treatment of SARS-CoV-2 infections with compounds that act by inhibiting viral replication is critical in order to achieve a significant reduction in viral titer and symptom improvement [18]. This reflects that as viral load reaches its peak, signifying the maximum viral replication and the highest amount of virus, the number of viral replications that can be inhibited by an antiviral agent begins to diminish, resulting in lower antiviral efficacy [19]. Hence, knowledge about the trajectory of the viral load, including the time of the peak, is pivotal for the timing of the treatment with direct-acting antiviral agents such as molnupiravir. Specifically, achieving a favorable treatment outcome may rely on initiating treatment with these agents before the viral load peak is reached [19,20,21]. 

Since the exact time at which infection with SARS-CoV-2 occurs in humans is often difficult to ascertain, the viral load peak is commonly reported with reference to symptom onset [22]. Symptoms of COVID-19 most often appear four to five days after infection in humans, albeit with shorter asymptomatic incubation periods for some SARS-CoV-2 variants [23]. Viral load peak occurs from symptom onset to a couple of days later in the upper respiratory tract of patients with COVID-19, and generally within one week after the onset of symptoms [24,25,26,27]. However, levels of SARS-CoV-2 in the upper respiratory tract have also been reported to peak before symptom onset [28,29]. In animal models of COVID-19, the disease course is generally shorter and more compressed than in humans, with first symptoms appearing two to three days after inoculation or even as early as one day after inoculation in the Roborovski dwarf hamster model, [30,31]. This model is also characterized by early viral peak titers occurring 1 to 2 days after inoculation [32]. Likewise, the viral load peaks early in the ferret, Syrian hamster, K18-hACE2 transgenic mouse, and macaque models of COVID-19, i.e., from 2 to 4 days after inoculation [31,32,33,34]. 

This report critically reviews studies that have investigated the efficacy of molnupiravir in animal models of COVID-19 and assesses whether the design of these studies might have influenced the efficacy of the agent and potentially contributed to inflated expectations about its anti-SARS-CoV-2 activity in humans. Specifically, we examined whether the design of the animal studies aligned with the use of the drug in humans for the treatment of COVID-19 in regard to dose, timing of treatment, and treatment duration. Based on this review, we suggest potential strategies to enhance the effectiveness of the agent in COVID-19.

## 2. Materials and Methods

### 2.1. Search Strategy

We searched PUBMED for studies examining the efficacy of molnupiravir in animal models using the search terms “Molnupiravir” AND “COVID-19” AND “Animal”. In subsequent searches the term “Animal” was replaced with “Mice”, “Ferret”, “Hamster” and “Macaque”. Based on this search strategy, we identified 18 relevant reports. As three of these reports evaluated the efficacy of molnupiravir using different animal models and design, a total of 22 studies were available for assessment.

### 2.2. Extracted Data 

We retrieved information regarding the experimental design and molnupiravir efficacy from the identified studies (Table 1). The experimental design included details such as the tested viral isolates, viral inoculation doses, dose levels of molnupiravir (mg/kg), time of initiation of treatment relative to inoculation, and treatment duration. Additionally, time from inoculation to viral load peak was determined. This determination was based on control animals, i.e., the animals administered vehicle instead of molnupiravir or other agent. Area under the curve from time zero to infinity (AUC_0-inf_) values for NHC after administration of a single oral dose of molnupiravir were also retrieved. Determination of efficacy of molnupiravir was based on decrease in SARS-CoV-2 load in tissues, swabs, nasal lavage, and bronchial lavage. Additionally, clinical signs, body weight loss, survival, lung pathology, and prevention of contact transmission served to evaluate antiviral efficacy. The viral load included infectious viruses (viral titer), viral RNA, and the virus-produced proteins. All assessments of the use of molnupiravir in animal models of COVID-19 were based on monotherapy findings of the drug. However, since the identified reports also included studies of molnupiravir in combination therapy, which have shown promise for improving treatment outcomes, we also briefly reviewed findings from these combination therapy studies.

### 2.3. Dose Calculations 

We normalized doses of molnupiravir in mg/kg according to body surface area in the animal species, which served as models of COVID-19, using previously reported conversion factors [35]. For humans, a standard body surface area was calculated using the DuBois formula [36], based on a height of 175 cm and a weight of 75 kg. This weight was used to account for the increased risk of severe illness in COVID-19 due to obesity [37], potentially resulting in higher body weights among COVID-19 patients, while also factoring in a lower body weight of females and differences in body weight across geographical regions.

**Table 1 viruses-15-02151-t001:** Doses, treatment timing, and efficacy of molnupiravir in animal models of COVID-19.

Animal Model ^1^	SARS-CoV-2 Isolate Used for Inoculation ^2^	Viral Dose ^3^	Oral Drug Dose in mg/kg, Bidaily	Oral Drug Dose in mg/m^2^, Bidaily ^4^	Start of Treatment Relative to Time of Infection	Treatment Duration ^5^	Efficacy of Molnupiravir ^6^	Study
SCID mouse	Beta	10^5^ TCID_50_	200	600	At inoculation	3 d	Reduced viral titers and viral RNA in lungs; improved lung pathology	Abdelnabi et al., 2022 [38]
K18-hACE2 mouse	Original type	5 MLD_50_	20	60	6 h after	5 d	Modest weight loss protection; improved clinical score; decreased viral RNA; viral titers in lungs largely unchanged; increased survival	Jeong et al., 2022 [39]
Lung-only mice	Original type	1–3 × 10^5^ PFU	500	1500	12 h before	2 d and 12 h	Markedly reduced lung viral titers with pre-inoculation treatment being most effective; lower viral antigen in lungs; improved lung pathology	Wahl et al., 2021 [40]
					24 h after	2 d		
					48 h after	2 d		
K18-hACE-2 mouse	Original type	300 FFU	50	150	2 h before	3 d and 20 h	Reduction in lung viral titers; lung pathology not improved	Stegmann et al., 2022 [41]
Syrian hamster	Original type, Alpha, Beta	10^5^ TCID_50_	200	1000	1 h before	4 d	Reduced viral titers and viral RNA in lungs; improved lung pathology; major weight increase for original viral type and Beta variant	Abdelnabi et al., 2021 [42]
Syrian hamster	Original type	2 × 10^6^ TCID_50_	75	375	1 h before	4 d	Lung virus titers not reduced by 75 mg/kg dose of molnupiravir but lowered by the higher doses; 150 mg/kg dose probably suboptimal for monotherapy	Abdelnabi et al., 2021 [43]
			150	750				
			200	1000				
Syrian hamster	Beta	10^4^ TCID_50_	150	750	At inoculation	4 d	Reduction in viral titers and virus RNA in lungs; lung pathology not significantly improved; no effect on body weight	Abdelnabi et al., 2022 [44]
Syrian hamster	Original type	10^6^ PFU	50	250	4 h before	5 d and 4 h	Full protection against weight loss with molnupiravir at 500 mg/kg and partial weight loss protection with the lower doses; lung viral titers decreased below detection limit with molnupiravir at 150 and 500 mg/kg; lung pathology improved	Bakowski et al., 2021 [45]
			150	750				
			500	2500				
Syrian hamster	Beta	10^4^ TCID_50_	150	750	1 h before	4 d	Reduced virus titers and virus RNA in lungs; improved lung pathology	Foo et al., 2022 [46]
Syrian hamster	Omicron	10^3^ PFU	500	2500	24 h after	3 d	Viral titers reduced in lungs but not in nasal turbinates	Uraki et al., 2022 [47]
Syrian hamster	Omicron	10^3^ PFU	500	2500	24 h after	3 d	Reduction in nasal turbinate viral titer on second day after inoculation; viable virus not detected in lungs during period treatment	Uraki et al., 2022 [48]
Syrian hamster	Original type	5 × 10^2^ TCID_50_	250	1250	12 h before	4 d	Reduced viral RNA, viral titers and viral antigen in lungs for treatments initiated both before and after inoculation; improved lung pathology; no effect on viral load in oral swabs	Rosenke et al., 2021 [49]
					2 h before	3 d and 14 h		
					12 h after	3 d		
Syrian hamster	Alpha, Beta, Delta, Omicron	10^3^ or 10^4^ TCID_50_	250	1250	12 h after	3 d	Reduced viral titers and viral antigen of all examined variants in lungs; reduced lung disease; no reduction in viral load in oral swabs except for Omicron variant	Rosenke et al., 2022 [50]
Syrian hamster	Original type	10^4^ TCID_50_	250	1250	24 h before	7 d and 12 h	Small decrease in nasal viral titer and in weight loss; improved lung pathology	Stegmann et al., 2022 [41]
Roborovski dwarf hamster	Delta, Gamma, Omicron	10^5^ PFU or 3 × 10^4^ PFU (Delta)	250	900	12 h after	11 d and 12 h	Prevented death by all SARS-CoV-2 types; reduced viral titers and viral RNA in lungs with larger reduction for Gamma variant; improved lung pathology for all virus variants	Lieber et al., 2022 [51]
Roborovski dwarf hamster	Omicron	10^4^ PFU	250	900	12 h after	4 d and 12 h (short) or 13 d and 12 h (long)	Short treatment reduced viral lung titer and prevented clinical signs, death, and viral rebound; long treatment prevented death, reduced lung viral titer but not viral RNA in lungs and trachea	Cox et al., 2023 [52]
Ferret	Alpha, Delta, Gamma, Omicron	10^5^ PFU	5	35	12 h after	3 d and 12 h	Titers in nasal lavages of all viral isolates below detection level 12 h after treatment start; blocked contact transmission; not all virus variants established productive infection	Lieber et al., 2022 [51]
Ferret	Original type	10^5^ PFU	5	35	12 h after	3 d and 12 h	Viral titers below detection limit in nasal lavages within 1 and 1.5 d for treatment started 12 and 36 h after inoculation, respectively; no contact transmission	Cox et al., 2021 [8]
			15	105	12 h after	3 d and 12 h		
			15	105	36 h after	2 d and 12 h		
Ferret	Original type	10^5^ PFU	1.25	8.75	12 h after	3 d and 12 h	All molnupiravir dose groups exhibited reduced viral titers and did not infect untreated contact animals	Cox et al., 2023 [52]
			2.5	17.5				
			5.0	35				
Ferret	Original type	10^5^ PFU	5	35	12 h before	6 d	No clinical signs, no infectious virus, and decreased viral RNA in nasal lavages and turbinates in prophylactically treated ferrets exposed to infected and untreated animals	Cox et al., 2023 [52]
Rhesus macaque	Original type	5.15 × 10^6^ or 6.08 × 10^6^ TCID_50_	75	900	At inoculation	7 d	Reduced nasal swab viral titers and viral RNA in bronchoalveolar lavages by administration of 250 mg/kg molnupiravir compared to dose of 75 mg/kg and vehicle; no effect on virus titers in bronchoalveolar lavage; lung pathology difficult to evaluate	Johnson et al., 2023 [9]
			250	3000				
Rhesus macaque	Delta	2 × 10^6^ TCID_50_	130 ^1^	1560	12 h after	3 d and 12 h	Reduced viral titers but not viral RNA in nasal and oral swabs; largely unchanged viral titers and viral RNA in lower airways; slightly milder disease course; less severe lung pathology	Rosenke et al., 2023 [10]

^1^ SCID mice: severe combined immunodeficiency mice; K18-hACE-2 mice: mice expressing human angiotensin-converting enzyme 2 under the control of the human keratin 18 (K18) promoter. ^2^ Infection is both inoculation and infection by contact transmission. Only WHO label of isolates are given. ^3^ TCID: 50% tissue culture infectious dose; PFU: plaque forming units; FFU: focus forming unit; MLD_50_: median lethal dose. ^4^ Body-surface-area-based doses. Conversion factors were based on standard body weights and surfaces [35]. For the dwarf hamster, a body weight of 25 g and a body surface of 0.007 m^2^ were used. The recommended oral human dose of molnupiravir is 800 mg bidaily (10.7 mg/kg), which corresponds to 400 mg/m^2^ bidaily for a standard person with a body weight of 75 kg, a height of 175 cm and a body surface area of 2.00 m^2^. ^5^ Consecutive days of treatment. In several studies, some animals were sacrificed before the end of treatment. These intermediate time points are not included in the table, which only provides the reported total duration of treatment. ^6^ In many studies, the viral load was determined prior to end of treatment. d = day(s) and h = hour(s).

### 2.4. Assessment of Data 

The experimental design in the reported animal studies of COVID-19 was evaluated with emphasis on the dose levels of molnupiravir, the timing of treatment with the drug, and treatment duration. Evaluation of efficacy was based on the reported treatment outcomes that varied across studies.

## 3. Results

### 3.1. Viral Doses and Variants

Different methods for quantification of viral inoculation doses were used in the reported animal studies, including median tissue culture infectious dose (TCID_50_), plaque forming units (PFUs), focus forming units (FFUs), and 50% mouse lethal dose (MLD_50_). Viral quantification was mostly conducted via TCID_50_ and PFUs, while the determination of FFUs and MLD_50_ were only used in a single study each. The doses ranged from 5 × 10^2^ to about 6 × 10^6^ TCID_50_ and from 10^3^ to 10^6^ PFUs; in addition, doses of 300 FFUs and 5 MLD_50_ were used. Several studies tested both the original types of SARS-CoV-2 and the Alpha, Beta, Delta, Gamma, and Omicron variants.

The use of various methodologies for viral quantification and distinct susceptibilities to SARS-CoV-2 across animal species complicates meaningful comparison of viral doses in the reported studies. Likewise, the pathogenicity was dependent on the SARS-CoV-2 variant, with the Delta variant being the most pathogenic of those tested in the dwarf hamster model [51]. Observations in this model also suggested that molnupiravir reduced lung titers of all the tested SARS-CoV-2 variants, with the efficacy being more pronounced for the Gamma variant than other variants [51]. Additionally, in the Syrian hamster model of COVID-19, the drug reduced the lung titers of the Alpha, Beta, and Delta variants, apparently possessing the highest activity against the latter, while the lung titer decrease observed for the Omicron variant did not reach statistical significance [50]. 

### 3.2. Dose Levels of Molnupiravir

The bidaily oral molnupiravir doses varied significantly across the examined studies ranging from 1.25 mg/kg in ferrets to 500 mg/kg in Syrian hamsters and lung-only mice. When we normalized the doses according to body surface areas in the individual species, they ranged from 6.25 to 3000 mg/m^2^ bidaily. In four studies based on mouse models of COVID-19 including K18-hACE-2 mice, SCID mice, and lung-only mice, molnupiravir doses ranged from 20 to 500 mg/kg [38,39,40,41]. The studies in SCID mice and lung-only mice both found significantly reduced pulmonary viral titers accompanied by improved lung pathology [38,40]. In the study based on the lung-only mouse model, molnupiravir at 500 mg/kg twice daily (1500 mg/m^2^ twice daily) was chosen, as this dose was assumed to provide intracellular 5′-triphosphate NHC levels similar to those in humans administered a dose of 1600 mg per day [40]. However, in humans, a daily dose of 1600 mg (800 mg twice daily) is equivalent to a body surface area-based dose of 400 mg/m^2^ twice daily, which is more than 3-fold lower than the 1500 mg/m^2^ dose twice daily in the lung-only mouse model. In the K18-hACE-2 mouse model, 20 mg/kg molnupiravir was selected based on a dose optimization study, and considered optimal for low-dose therapeutic efficacy [39].

Studies of the efficacy of molnupiravir in Syrian hamsters inoculated with SARS-CoV-2 were carried out with doses ranging from 50 to 500 mg/kg twice daily [41,42,43,44,45,46,47,48,49,50]. Specifically, a dose–response study suggested that a 150 mg/kg dose of molnupiravir twice daily was suboptimal, and that a dose of at least 200 mg/kg twice daily was necessary to reduce pulmonary viral titers to levels near the detection limit [43]. However, in another study using the Syrian hamster model, the administration of molnupiravir at a dose of 150 mg/kg twice daily not only reduced pulmonary viral titers to below the detection limit, but also provided partial protection against weight loss, demonstrating a weight loss that was negligible with a higher molnupiravir dose of 500 mg/kg twice daily [45]. 

Some studies in the Syrian hamster and dwarf hamster models, selected molnupiravir doses based on findings from other animal models of COVID-19 [48,51]. For example, the molnupiravir dose of 500 mg/kg twice daily was administered to Syrian hamsters, as this dose was previously demonstrated to be efficacious in the lung-only mouse model of COVID-19 [48]. Moreover, the choice of a bidaily dose of 250 mg/kg molnupiravir for dwarf hamsters was motivated by their heightened metabolic activity compared to ferrets, which were administered 5 mg/kg bidaily, and aligned with the bidaily doses employed in Syrian hamsters and lung-only mice of 250 and 500 mg/kg, respectively [51]. In dwarf hamsters inoculated with SARS-CoV-2, treatment with 250 mg/kg molnupiravir twice daily prevented death and reduced lung viral titer, but not viral RNA in lungs and trachea, and animals that discontinued treatment after five days survived without clinical signs and viral rebound [52]. Ferrets received molnupiravir in bidaily doses ranging from 1.5 to 15 mg/kg [8,51,52]. These doses were the lowest among the doses used in the animal models, both when expressed as mg/kg and mg/m^2^. However, they effectively eliminated SARS-CoV-2 from nasal passages in the ferrets after 0.5 to 1.5 days of treatment and prevented contact transmission [8,51,52].

Rhesus macaque monkeys were administered molnupiravir at doses of 75, 130, and 250 mg/kg twice daily [9,10], which were equal to 900, 1560, and 3000 mg/m^2^, respectively. The 75 and 250 mg/kg twice-daily dose levels were selected based on previous knowledge about the plasma exposure of molnupiravir in macaque monkeys and the in vitro anti-SARS-CoV-2 activity of the agent [9]. The 130 mg/kg twice-daily dose of molnupiravir was reported to be allometrically derived from the 800 mg twice-daily dose recommended for treatment of COVID-19, although the calculation underlying this dose conversion was not specified [10]. The molnupiravir doses of 75 and 130 mg/kg, administered twice daily, appeared to be therapeutically suboptimal, with varying efficacy across treatment outcomes in the macaque model [9,10]. Additionally, the dose of 250 mg/kg twice daily did not consistently reduce viral RNA and titers in this model [9]. Importantly, assessing the efficacy of molnupiravir in the macaque model posed challenges due to mild clinical disease after inoculation with SARS-CoV-2, but the agent successfully reduced the viral load in the airways [9,10]. 

Most of the molnupiravir doses used in the reported animal models of COVID-19, expressed in both mg/kg and mg/m^2^, exceeded the corresponding doses of 10.7 mg/kg and of 400 mg/m^2^ twice daily, respectively, that were derived from the recommended human dose of 800 mg twice daily. Notably, in over half of the animal studies, including the two studies performed in macaques, the calculated body-surface-area-based doses of molnupiravir exceeded the human equivalent dose of 400 mg/m^2^ by a factor of two or more. 

Presently, AUC_0-inf_ values of NHC after a single oral dose administration of molnupiravir have been reported for dwarf hamsters, macaques, and ferrets [52]. In dwarf hamsters, this value was 376 h × ng/mL after the administration of 250 mg/kg, while a value of 25,299 h × ng/mL was reported after the administration of 130 mg/kg in macaques. The AUC_0-inf_ values in ferrets were 3421, 7569 and 18,793 h × ng/mL after administration of single oral doses of molnupiravir of 4, 7 and 20 mg/kg, respectively [52]. In humans, the AUC_0-inf_ of NHC with a single oral dose of 800 mg molnupiravir was determined at 8740 h × ng/mL [52,53]. Importantly, the 250 mg/kg dose in dwarf hamsters and the lowest dose in ferrets of 4 mg/kg, which both likely have efficacy against experimental infection with SARS-CoV-2, produced lower AUC_0-inf_ values than those determined after the administration of a single oral dose of 800 mg in humans, with the AUC_0-inf_ value in dwarf hamsters being more than 23-fold lower than that in humans. Since NHC exposure values were not available for mice and Syrian hamsters, a comparison with the NHC exposure in humans administered an 800 mg dose of molnupiravir was not possible.

Several of the animal studies aimed to assess the effect of combining molnupiravir with another agent including an antiviral. These studies used a suboptimal dose of molnupiravir to compare its effect in monotherapy with that observed in combination therapies, thereby facilitating the detection of potential synergistic effects [10,39,41,43,44,46]. For example, a suboptimal molnupiravir dose of 150 mg/kg twice daily was used in Syrian hamsters [43,44,46]. Importantly, the combination of this dose of molnupiravir with favipiravir or GS-441524 (the nucleobase of remdesivir), exhibited potent antiviral activity in Syrian hamsters [43,44]. Combining doses of 250 mg/kg of molnupiravir and 10 mg/kg of teriflunomide twice daily in the Syrian hamster model also suggested the superiority of the combination over each of the compounds in monotherapy, with molnupiravir alone reducing nasal lavage titers by less than 1.5 log_10_ [41]. 

### 3.3. Time from Inoculation to Viral Load Peak 

Due to the lack of a sufficient number of consecutive samplings, it was only possible to determine the time of the viral load peak relative to that of inoculation with SARS-CoV-2 in some of the animal studies. However, in the lung-only mouse model, the viral titer was reported to peak two days after the inoculation [40]. Viral load peaked in nasal turbinates on the second day after inoculation and one to two days later in the lungs of Syrian hamsters, with these levels remaining high for several days [48]. Likewise, the RNA levels of the Alpha, Beta, Delta, and Omicron variants in oral swabs collected from Syrian hamsters appeared to plateau at a constantly high level from 2 to 3.5 days after inoculation [50]. Also, viral titers in nasal washings from Syrian hamsters decreased from two to four days after inoculation, consistent with virus load peak occurring within the first couple of days after the inoculation [41]. By contrast, virus load peaked in the lungs of dwarf hamsters as early as 12 to 24 h after inoculation [51]. Likewise, the replication of SARS-CoV-2 appeared to reach an early maximum in the upper respiratory tract of ferrets, where the viral load has been found to peak in nasal washings at 24 to 36 h after inoculation [8]. This aligns with other findings in the studies under assessment, which suggested that viral load peak in ferrets occurred one to three days after inoculation, depending upon the virus variant [51]. Moreover, findings in macaques suggested that the viral load in nasal swabs peaked about two days after inoculation with SARS-CoV-2 [9,10].

### 3.4. Time to Initiation of Treatment

Treatment with molnupiravir was initiated at various time points in the animal studies, spanning from 24 h before to 48 h after SARS-CoV-2 inoculation, with most treatments being initiated before or concurrently with the viral inoculation. Several studies permitted the comparison of the outcome of treatments initiated at different time points relative to the time of inoculation, thereby providing the basis for the determination of a therapeutic time window, i.e., the time interval where the treatment is likely to be most effective. This included a study in the lung-only mouse model using 500 mg/kg molnupiravir twice daily, where the initiation of this treatment 12 h before inoculation led to lower lung viral titers and improved lung pathology compared to treatments initiated 24 and 48 h after inoculation, with the lowest therapeutic effect observed in the animals initiating treatment 48 h after inoculation [40]. Likewise, a study using the Syrian hamster model of COVID-19 reported that bidaily treatment with molnupiravir at 250 mg/kg that was initiated 12 h prior to inoculation resulted in significantly lower viral titer, RNA, and antigen in the lungs than treatments initiated 2 h before or 12 h after inoculation [49]. In line with these findings, lower nasal titers of SARS-CoV-2 were reported in ferrets that initiated treatment with molnupiravir 12 h after inoculation compared to those initiating molnupiravir treatment 36 h after inoculation [8].

Although molnupiravir treatment was initiated before or simultaneously with inoculation in most studies, several studies exclusively initiated the treatment after inoculation. This included studies in mice, Syrian hamsters, dwarf hamsters, ferrets, and macaques, which started treatment at 6, 12, and 24 h after viral inoculation [8,10,39,47,48,50,51,52]. For example, the treatment of K-18 hACE2 mice with molnupiravir at 20 mg/kg twice daily was initiated 6 h after inoculation and provided only modest protection against weight loss, but improved the clinical score and decreased mortality significantly [39]. Additionally, Syrian hamsters initiating treatment with doses of molnupiravir at 250 or 500 mg/kg twice daily at 12 and 24 h post-inoculation exhibited significantly decreased lung viral titers for both the original and variant strains of SARS-CoV-2, while viral reductions in nasal turbinates and oral swabs were not consistent over the course of the experiments [47,48,50]. Finally, the treatment of experimentally infected macaques with a 130 mg/kg twice-daily molnupiravir dose initiated 12 h after inoculation reduced clinical signs and also lowered viral titers and antigen levels, but did not decrease levels of viral RNA in the upper and lower airways [10].

In general, findings from the animal studies where treatment was exclusively initiated post-inoculation indicated that molnupiravir was effective in eliminating SARS-CoV-2 or preventing transmission, although its efficacy differed across outcome measures. It is, however, noteworthy that several of these studies used molnupiravir doses at the upper end of those administered in the reported animal models, including the 500 mg/kg twice daily in mice and Syrian hamsters [1,47,48].

### 3.5. Duration of Treatment

The duration of treatment varied in the animal studies and ranged from 2 days to almost 13.5 days with a median of 4 days, reflecting differences in study design. The study with the treatment duration of only 2 days was performed in the lung-only mouse model and used the highest molnupiravir doses among all the reported animal studies, namely, 500 mg/kg [40]. The longest treatment durations were 11.5 and 13.5 days in two studies with Roborovski dwarf hamsters [51,52], followed by 7 days in studies using macaques [9] and Syrian hamsters [41]. Hence, these treatment durations exceeded the recommended five-day course for the treatment of COVID-19. 

## 4. Discussion

In the present work, we reviewed studies examining the efficacy of molnupiravir in various animal species and collected detailed information about their design. Based on this information, we assessed how well these animal studies aligned with the clinical trials of the efficacy of molnupiravir in COVID-19, primarily focusing on discrepancies in drug dose, timing of treatment, and treatment duration.

### 4.1. Viral Strains and Doses

The inoculation dose and variant of SARS-CoV-2 may impact the disease course and severity, potentially influencing the efficacy of antiviral treatment [54]. The assessment of antiviral treatment efficacy in animal models often involves the use of viral doses of a magnitude sufficient to induce noticeable clinical signs, facilitate the detection of treatment effects, and mimic specific features of COVID-19 [54,55]. However, this empirical and somewhat arbitrary approach may potentially impact the outcomes of antiviral treatment, and when combined with varying susceptibility to SARS-CoV-2 in various animal species, it complicates the comparison of viral doses used across animal models. 

Strains of SARS-CoV-2 vary in several aspects. This includes variation in replicative efficacy, with the Omicron variant reported to replicate faster than other SARS-CoV-2 variants in the bronchi but slower in the lung parenchyma [56]. This is relevant as the initiation of molnupiravir treatment promptly during the early stages of infections with rapidly replicating SARS-CoV-2 variants could be more effective than its use in infections caused by variants with slower replication, despite the potential of rapidly replicating variants for generation of high viral loads [19]. On the other hand, molnupiravir was found to be highly effective against the Alpha, Beta, Omicron, Gamma, and Delta variants in the dwarf hamster model, although there were differences in disease severity and time to onset of symptoms [51]. This suggests that a differential effect of molnupiravir across the examined variants of SARS-CoV-2 was not a major determinant of the observed treatment outcomes.

### 4.2. Dose Levels of Molnupiravir

There was large variation in the molnupiravir doses used across animal models in the assessed studies and a lack of clarity in the rationale behind the selection of these doses in some of the studies. Based on doses in mg/kg and in mg/m^2^, most of the animal doses exceeded the recommended 800 mg bidaily dose in humans. Body-surface-area-based scaling enables the conversion of drug doses from one species to another, and is often used for the calculation of the human dose when a drug proceeds from animal studies and the preclinical development phase to the first trials in humans, but it requires similar pharmacokinetics and pharmacodynamics in humans and the animal species used [57,58,59]. If these requirements are not met, the method may underestimate the effective dose in humans [57]. Hence, achieving a more accurate understanding of the relationships between drug dose, exposure, and efficacy across species may require sophisticated modeling based on species-specific pharmacological parameters, as demonstrated in the studies of remdesivir and GS-441524 [60,61]. 

Major differences in the pharmacokinetics of molnupiravir appeared to exist across species. Notably, in dwarf hamsters, a single oral molnupiravir dose of 250 mg/kg produced a more than 67-fold lower NHC exposure in plasma than a single oral dose of 130 mg/kg in macaques [52]. Moreover, single oral doses of molnupiravir at 800 mg in humans and 13 mg/kg in ferrets produced NHC plasma AUC_0-inf_ values that were comparable but about 3-fold lower than that in macaques and more than 20-fold higher than the value in dwarf hamsters. These findings, based on NHC plasma exposure, suggest that the 250 mg/kg dose in dwarf hamsters would underestimate the efficacy of an 800 mg dose of molnupiravir for the treatment of COVID-19, while the 13 mg/kg ferret dose aligns with the human dose. Furthermore, macaques display high NHC levels in plasma following the oral administration of molnupiravir, suggesting that the 130 mg/kg dose used in these animals would exhibit significantly greater activity against SARS-CoV-2 than the recommended dose for human treatment. This is interesting because macaques, among the species used to assess the efficacy of molnupiravir, are the closest to humans. Moreover, along with other nonhuman primates, macaques have played a significant role in evaluating the efficacy of molnupiravir against COVID-19 and developing novel therapeutics for the virus [33]. 

Increasing the dose of molnupiravir above the recommended 800 mg twice daily could be a strategy to enhance its efficacy against COVID-19. This seems justified based on a systematic review of six published studies and data from 18 ongoing trials, which suggested that molnupiravir is generally safe and well-tolerated [62]. Also, a more recent systematic review and meta-analysis of nine randomized clinical trials comprising about 30,000 patients found molnupiravir to be without serious adverse events, although there was an indication of increased mortality in one of these trials, which was a small trial of inpatients [63]. However, before considering the use of higher doses of molnupiravir in COVID-19, further investigations would be imperative to clarify whether the drug increases inpatient mortality [63] or gives rise to other serious adverse events, as suggested by the examination of post-marketing pharmacovigilance data [64]. Moreover, the assessment of the risk of mutagenicity and embryotoxicity of the drug in humans would be crucial [65,66], along with considerations of its potential for the generation of novel SARS-CoV-2 variants, which may result from the induction of transitions in the viral genome characteristic of the mechanism of action of the drug [67]. 

Compared with systemic administration, the inhalation of an antiviral may be more advantageous for the treatment of viral infections in the airways. Notably, higher concentrations of the active nucleotide triphosphate metabolite of remdesivir were detected in the lungs of BALB/c mice with the pulmonary administration of liposome-encapsulated remdesivir compared to the intravenous injection of the agent [68]. Additionally, in the African green monkey model of SARS-CoV-2 infection, the inhalation of remdesivir led to a reduction in viral RNA in bronchoalveolar lavage fluid and respiratory tract tissues [69]. Along this line, recent research has examined the pharmacokinetics, safety, and tolerability of inhaled remdesivir in humans, suggesting that this route of administration holds clinical promise [70]. 

Combining molnupiravir with another agent with the aim of achieving a potentiated or even synergistic antiviral effect may allow for the use of lower doses of both agents in the combination, thereby reducing the risk of undesired effects [43,44]. The findings with molnupiravir in combination with favipiravir, GS-441524, the protease inhibitor nirmatrelvir, and inhibitors of dihydroorotate dehydrogenase in animal models of COVID-19 [10,39,41,43,44] suggest a potential for clinical development. However, data are still too limited to determine which combination with molnupiravir holds the most promise for clinical use. It is noteworthy that GS-441524 targets the viral RNA-dependent RNA polymerase, acting as an RNA chain terminator [71], while molnupiravir, and likely favipiravir as well, do not block this polymerase but instead introduce lethal mutations in the viral progeny genome [72,73,74,75]. At present, clinical trial data on the efficacy and safety of molnupiravir in combination therapy for the treatment of mild COVID-19 are lacking, and there are no registered studies of molnupiravir in antiviral combination therapy on ClinicalTrials.gov [62,76]. However, the successful off-label use of a combination of molnupiravir and nirmatrelvir/ritonavir has been reported in a single difficult-to-treat COVID-19 patient [77].

### 4.3. Time from Infection to Viral Load Peak 

We collected data on the time from inoculation to the peak of viral load from molnupiravir studies conducted on animal models of COVID-19, as this interval is considered optimal for antiviral treatment. The intervals in these studies appeared to align with observations in the studies that established and characterized the used animal models of COVID-19 and showed that viral load typically peaked between one to four days after inoculation, with the earliest occurrence of the peak observed in the dwarf hamster model [32,34,40,78,79,80]. Hence, the infections in the animal models of COVID-19 used for the examination of the efficacy of molnupiravir appeared to have followed a typical course. Importantly, while the time from infection to the viral load peak is considered a crucial parameter for the timing of effective antiviral treatment, sophisticated within-host modeling of the viral dynamics may provide more accurate information about the course of the infection, as well as informing treatments [19,24,81,82]. Such modeling also holds promise for facilitating the translation of results relevant to the treatment of COVID-19.

### 4.4. Time to Initiation of Treatment 

Since molnupiravir treatment was initiated within two days after infection with SARS-CoV-2 in most of the reported animal studies, exposure to the drug likely commenced before the viral load peak, although in dwarf hamsters and ferrets, the viral load peak may have occurred earlier in the infection course. In the MOVe-OUT study, where molnupiravir was deemed clinically effective, about half of the patients initiated treatment within three days or less from symptom onset, while treatment was started four to five days after symptom onset in the remainder of the patients [11]. The median time from symptom onset to treatment initiation was three to five days in the subsequent PANORAMIC study, which failed to demonstrate a reduced risk of death and hospitalization in patients treated with molnupiravir [14]. If the viral load peaks within a few days after symptom onset or even earlier in some of the humans infected with SARS-CoV-2, treatment initiation in MOVe-OUT and PANORAMIC would have come belatedly, resulting in a decreased overall efficacy of molnupiravir. Indeed, a retrospective observational study found that initiating molnupiravir treatment three days or more after the diagnosis of COVID-19 resulted in slower viral clearance compared to earlier initiation, highlighting the importance of timely treatment with the drug [83]. In the animal studies, treatment with molnupiravir appeared to have been initiated even earlier relative to the viral load peak. This early initiation may have contributed to higher efficacy and the raised expectations for outcomes in human clinical trials with the agent. However, comparison of the duration of the therapeutic window of molnupiravir in COVID-19 and animal models of the disease is not straightforward due to the typically condensed disease course with an earlier viral load peak in the animal models. 

### 4.5. Duration of Treatment

The treatment duration for humans with molnupiravir is limited to five days due to safety concerns, whereas treatment was extended in several animal studies. Hence, the generally more condensed disease course in animal models of COVID-19 compared to humans [30,31] did not necessarily translate into proportionately shorter treatment durations with molnupiravir in the reported studies. This is of potential importance since the extended duration of treatment with molnupiravir in some of the animal studies may have resulted in enhanced antiviral efficacy. This again underscores the challenges associated with the application of animal models of COVID-19 and raises the question of whether the treatment lengths selected in all these models aligned with that recommended for treating COVID-19. Finally, it is worth noting that when treatment with molnupiravir, administered twice daily, is discontinued, NHC exposure levels may exhibit antiviral activity for several hours after the treatment discontinuation. This effect could have a larger impact on antiviral efficacy in animal studies with short treatment durations.

### 4.6. Potential Confounding Factors

Several factors may have influenced the outcomes of studies assessing the efficacy of molnupiravir in animal models of COVID-19. These factors include the use of animal species and strains as models of COVID-19 with disease characteristics and pharmacokinetics of molnupiravir distinct from those in humans. For example, K18-hACE2 mice may develop a lethal brain infection after SARS-CoV-2 inoculation, i.e., a feature not characteristic of COVID-19, and they do not recapitulate other features of the disease, such as multi-organ failure [84,85]. Additionally, the preferential use of young, resilient animals with disease courses that do not mimic those observed in older humans may have contributed to enhanced molnupiravir efficacy and confounded treatment outcomes in the animal studies. Biological sex represents another potential confounding factor in the animal studies. Importantly, molnupiravir has been found to be efficient against the Omicron variant of SARS-CoV-2 in male but not in female dwarf hamsters [51]. Moreover, after low-dose inoculation of K18-hACE2 mice with SARS-CoV-2, 60% of the female mice survived, while all those with the male gender succumbed to the infection [84]. This aligns with the disease in humans, where sex has been suggested to affect susceptibility to infection, immunopathogenesis, and other elements in the pathogenesis of COVID-19, thereby contributing to the mechanisms underlying sex-related differences in disease outcomes, including a higher fatality rate in males [86]. Furthermore, the small sizes of the experimental groups in the animal studies increase the risk of random events or fluctuations in observed outcomes being mistaken for treatment effects. Finally, sample size calculations were frequently not reported in the animal studies. 

Assessing the efficacy of a drug in clinical trials and real-world treatment is evidently much more complex than studies in animals [87,88,89,90]. This complexity can be attributed to multiple factors, particularly patient heterogeneity resulting from variations in age, gender, comorbidity, ethnicity, genetic background, and environmental conditions, including lifestyle factors [88,90,91]. Also, ethical considerations and informed consent impose limitations in clinical trials that are not present in animal studies. Furthermore, the emergence of variants potentially differing in antiviral susceptibility adds an extra layer of complexity to clinical trials assessing the efficacy of antivirals. It is also worth noting that while treatment timing may not be critical for some classes of drugs, the direct-acting antivirals may have a short therapeutic time window that can be accounted for in animal studies but not in clinical trials. 

## 5. Conclusions and Future Perspectives 

Potentially significant limitations in the studies assessing the efficacy of molnupiravir in the animal models of COVID-19 pertain to the dose levels, treatment timing, and duration of treatment. The dose levels of molnupiravir in some animal studies may have been excessively high compared to the recommended human dose, leading to significantly higher drug exposures and antiviral activities. Similarly, prophylactic treatment with molnupiravir, and treatment initiated shortly after viral inoculation, i.e., prior to viral load peak, in the animal models, likely increased its treatment effect. Furthermore, in many animal studies, the treatment duration appeared to be disproportionately longer than the recommended treatment duration in humans, given the typically short and condensed disease course in animal models of COVID-19, with several studies exceeding the human treatment duration. Hence, we posit that design elements in the animal studies assessing the efficacy of molnupiravir contributed to the overestimation of the efficacy of the agent, and thus inflated the expectations for its application in COVID-19. 

Our study has potential implications for future research and for clinical practice in the longer term. First, we suggest implementing a degree of standardization in conducting and reporting antiviral testing in animal models of COVID-19. This may facilitate comparisons across studies and potentially bridge the gap between animal models and human clinical trials, thereby increasing translatability. Second, we speculate that addressing the limitations in the animal studies might provide strategies for enhancing the efficacy of molnupiravir against COVID-19. Therefore, we propose considering the use of higher doses than the currently recommended human oral dose of 800 mg twice daily, administering molnupiravir or NHC via inhalation, initiating treatment early during the asymptomatic phase, and combining molnupiravir with other antivirals. Importantly, these approaches should all be supported by rigorous safety data.

## Data Availability

The data presented in the report and used for calculations were derived from publicly available studies.

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
