# Peer review of "Molnupiravir Revisited—Critical Assessment of Studies in Animal Models of COVID-19"

_viruses, 2023, doi:10.3390/v15112151_

Round 1

Reviewer 1 Report

Comments and Suggestions for Authors

The study assessed the design elements of animal studies investigating the efficacy of molnupiravir in COVID-19 and found certain factors that may have biased the results and inflated expectations for its effectiveness. Considering these design elements, the report suggests potential avenues to enhance the clinical efficacy of molnupiravir for COVID-19 treatment. These include considering dose increment, early treatment, administration by inhalation, and exploring the use of molnupiravir in combination with other antiviral therapies.

I recommend accepting this article after MINOR REVISIONS.

1. There are several recent animal models reports and clinical studies which are not included in this manuscript. Authors should update the manuscript by adding recent studies. DOI: 10.1093/cid/ciad504; DOI: 10.1038/s41467-023-40556-8; DOI: 10.1093/infdis/jiad195.

2. There is a lack of recent literature citations. The authors should enrich the related articles published between 2020 and 2023. For example, in lines 37-38, “Molnupiravir is the isopropylester derivative of the nucleoside analog β-D-N4-hydroxycytidine (NHC). (DOI: 10.1002/jmv.27517).”

3. In “4. Conclusion and future perspectives” section. Please discuss the following “While these suggestions may offer hope for improving the effectiveness of molnupiravir, it is crucial to emphasize the need for rigorous safety data alongside any new clinical developments. Ensuring the safety profile of any potential treatment is essential before considering its widespread use in treating COVID-19 or any other medical condition. It is also important to note that subsequent clinical studies did not confirm the initial efficacy observed in animal models, leading to the rejection of molnupiravir for permanent market authorization in many countries. Therefore, further research and clinical trials are necessary to determine the true efficacy and safety of molnupiravir in treating COVID-19.”

4. “Covid-19” please change with “COVID-19”.

Comments on the Quality of English Language

Minor editing

Reviewer 2 Report

Comments and Suggestions for Authors

The manuscript “Molnupiravir revisited – critical assessment of studies in animal models of Covid-19” summarizes experimental preclinical approaches to assessment of anti-viral activity of molnupiravir against SARS-CoV-2 virus on various animal models. After the analysis of accessible data, the conclusion is made that the course and peculiarities of virus-caused pathological process in animals often does not correspond to human ones. The same is shown to be true for treatment doses and schedules.

The manuscript is highly important for both theoretical virologists and specialists in areas of drug development and pre-clinical studies. Its publication, in my opinion, is highly desirable. The manuscript is well written and suggests high professional level of authors.

Minor corrections should be done prior to publication.

Line 67-68. It’s better to avoid mentioning viral particles. Although this is partially true, this parameter itself does not reflect the severity of disease; and the infectious viral titer depends on the amount of complete, infectious particles, not all particles together. I would use the term “viral titer”, “infectious virus”, etc.

Line 117. Change to read “bronchial lavage”

Everywhere within the text. Please spell “COVID-19” in capitals.

Line 204. Change to read “efficacy of molnupiravir”

Reviewer 3 Report

Comments and Suggestions for Authors

General feed back

This narrative review examined the efficacy oMolnupiravir in animal studies.

Adjusting for body surface area, in 50% studies  Molnupiravirdoses in animal studies more than doubled human doses in clinical trials. Moreover, differently from human clinical studies, in animal models Molnupiravir was promptly administered after SARS-CoV-2 inoculation.

• Lines 175-180In a recent observational study adjusting for an number of covariates including days since symptoms onset and treatment start, vaccination and Chalrson index 0 high risk COVID-19 patients treated by Molnupiravir were admitted to hospital, proving that Mulnupiravir was at least as good as Paxlovid [PMID: 37242504]. Moreover, adjusting for number swab tests undertaken until 

First negative test result, the efficacy of Molnupiravir was similar to Paxlovid in reducing SARS-CoV-2 shedding time from the nasal cavity [PMID: 37242504].

• Lines 204-206: This sentence lacks the verb • Lines 224-226: “!Therefore, we propose achieving enhanced efficacy 224 of molnupiravir in Covid-19 by using higher doses than the currently recommended oral 225 dose of 800 mg twice daily”… in humans? Please specify

• PRISMA; ROBINS AND GRADE criteria were considered?

Reviewer 4 Report

Comments and Suggestions for Authors There is really one point. It is not possible to use allometric scaling to work out dose rates for this drug in different species The flaw of ALL THE STUDIES cited, and this metanalysis - is that they do not determine peak and trough levels of molnupiravir in each species. If you do not do that - you cannot make any sense of the data. Comments on the Quality of English Language Minor

Reviewer 5 Report

Comments and Suggestions for Authors

Overall, the manuscript provides a critical assessment of the existing literature on molnupiravir's efficacy in animal models of Covid-19 and its implications for human studies. While the paper presents some valuable insights, there are several areas that require attention and improvement:

  1. Clarity and Organization: The manuscript's structure is somewhat confusing, with the introduction and methods section missing, and the flow of the argument not well-structured. It's important to clearly define the research question, methodology, and objectives at the beginning.
  2. Referencing: The manuscript frequently mentions "we" and "our findings" without clear reference to the methodology used in this critical assessment.
  3. Citation of Studies: The manuscript refers to "19 studies published in 17 reports," but the reader is left wondering which specific studies and reports are being discussed. Providing a list of these studies or detailed citations is essential for transparency and replicability.
  4. Bias and Assumptions: The manuscript seems to make strong claims about bias in animal studies favoring molnupiravir without providing concrete evidence or a rigorous analysis of each study's design and methodology. A more detailed and systematic review of these studies is necessary to support these claims.
  5. Alternative Explanations: The manuscript presents a hypothesis that high doses, prophylactic treatment, and longer treatment duration may have favored molnupiravir in animal models. However, it doesn't adequately address alternative explanations or potential confounding factors that could have influenced the outcomes in these studies.
  6. Human Clinical Trials: While the manuscript criticizes animal studies for not aligning with human trials, it should also acknowledge the complexities of human trials, including patient heterogeneity, ethical considerations, and the real-world challenges of administering treatments.
  7. Dose Recommendations: The suggestion to use higher doses than recommended for humans without solid evidence could be risky and may require further investigation and safety assessments. It's essential to balance the potential benefits with safety concerns.
  8. Inhalation Administration: The idea of administering molnupiravir via inhalation is intriguing but lacks supporting evidence or discussion of potential challenges, such as patient compliance, delivery mechanisms, or side effects.
  9. Combination Therapy: While combining molnupiravir with other antivirals is suggested for enhanced efficacy, the manuscript should elaborate on which specific antivirals and their mechanisms of action would be most promising in combination therapy.
  10. Safety Concerns: The manuscript briefly mentions safety concerns but does not delve into the potential risks or how they should be addressed. A more comprehensive discussion of safety is necessary.
  11. Conclusion: The conclusion should summarize the key findings and recommendations more concisely and explicitly state the implications for future research and clinical practice.

In conclusion, while the manuscript raises important points about the design of animal studies and their implications for human trials of molnupiravir, it lacks depth, specificity, and a clear methodology. A more systematic and evidence-based approach is needed to support the claims and recommendations made in the manuscript. Additionally, improving the organization and clarity of the manuscript will enhance its readability and impact.

    1.  
Comments on the Quality of English Language

Final proof reading needed as there are some typos
